# Genetic architecture of a light-temperature coincidence detector

Adam Seluzicki [1,2] ✉ & Joanne Chory [1,2,3]

Light and temperature variations are inescapable in nature. These signals provide daily and seasonal information, guiding life history determinations across many taxa. Here we show that signals from the PHOTOTROPIN2 (PHOT2) blue photoreceptor combine with low temperature information to control flowering. Plants lacking PHOT2 flower later than controls when grown in low ambient temperature. This phenotype requires blue light, is blocked by removal of NON-PHOTOTROPIC HYPOCOTYL 3 (NPH3), and is recapitulated by removing the transcription factor CAMTA2. PHOT2 and CAMTA2 show non-additive genetic interactions in phenotype and gene expression. Network-based co-expression analysis indicates system-level control of key growth modules by PHOT2 and CAMTA2. CAMTA2 is required for low temperature up-regulation of *EHB1*, a known NPH3-interacting protein, providing a potential mechanism of temperature information input to the PHOT-NPH3 blue light signaling system. Together these data describe the genetic architecture of environmental signal integration in this blue light-low temperature coincidence detection module.

Daily and seasonal cycles of light and temperature provide information that organisms transform into modulation signals, tuning physiology and development to the environment[1–7]. Light quality, intensity, duration, and direction are sensed in parallel by a suite of photoreceptor proteins[8]. Temperature sensing involves many systems, including calcium currents, membrane fluidity, RNA secondary structure, histone dynamics, and the photoreceptors themselves[9–15]. Light and temperature are processed by plants simultaneously, with the combination of light and temperature conditions providing more information than either stimulus alone. Physiological and developmental functions that demonstrate differential sensitivity to particular combinations of environmental light and temperature are of interest to understand mechanisms of processing environmental information.

PHOTOTROPINs (PHOTs) are blue-light-activated kinases, localized to the plasma membrane in complexes with accessory signaling factors[16]. PHOT1 is more sensitive than PHOT2[14,17]. Both participate in phototropic bending, chloroplast movement, leaf flattening, and stomatal opening[18–21]. Several members of the NON-PHOTOTROPIC

HYPOCOTYL 3 / ROOT PHOTOTROPISM 2 (NPH3 / RPT2)-Like (NRL) family of BROAD-TRAMTRACK-BRICK-A-BRACK (BTB)-domain signaling proteins participate in PHOT-dependent processes. NPH3 and RPT2 contribute to phototropic bending, RPT2 and NCH1 function in chloroplast movement, and all three are involved in controlling leaf curvature[22]. While the other photoreceptor families have well-described links to regulation of flowering, there have been few indications that PHOT signaling components contribute to this developmental transition[18].

CALMODULIN-BINDING TRANSCRIPTIONAL ACTIVATOR (CAMTA) transcription factors are widespread in eukaryotes. They are involved in neural and cardiac development and signaling in animals, and regulate immunity, drought, salt, and hormone responses in plants[23–28]. In *Arabidopsis*, CAMTA3 is the best studied due to its role in regulating immunity in response to low temperature, acting redundantly with CAMTA1 and 2[24,25,29]. There are conflicting reports regarding a possible interaction between CAMTA3 and the NRL protein NCH1[26,30]. Temperature regulation of CAMTA3 function has been mapped to the

[1]Howard Hughes Medical Institute, Salk Institute for Biological Studies, La Jolla, CA, USA. [2]Plant Molecular and Cellular Biology Laboratory, Salk Institute for Biological Studies, 10010 North Torrey Pines Road, La Jolla, CA, USA. [3]Deceased: Joanne Chory. This manuscript is dedicated to the memory of Dr. Joanne Chory. ✉e-mail: aseluzicki@salk.edu

DNA-binding domain[31]. Other CAMTAs have not been studied in as much detail. Outside of calcium, this family of transcription factors has not been directly linked to the regulation of known signaling pathways in plants.

Here, we examine the PHOTOTROPIN contribution to flowering in the context of environmental light-dark cycles and ambient temperature. We show that NPH3 functions at the center of a coincidence detector, integrating blue light intensity from PHOT2 and low ambient temperature from CAMTA2. We provide genetic and molecular evidence that PHOT2 and CAMTA2 work together to fine-tune flowering and regulate highly overlapping gene sets. Network co-expression analysis shows system-level misregulation of fundamental growth control modules in *phot2* and *camta2* mutants. Low temperature- and CAMTA2-dependent expression of the NPH3-interacting protein EHB1 is a likely point of temperature information input to the PHOT-NPH3 light signaling pathway. Thus, PHOT2, CAMTA2, and NPH3 function together to integrate light and temperature information, buffering development against environmental conditions.

## Results

### PHOT2 and NPH3 control coordinated light and temperature sensitivity

We analyzed the relationship between PHOTOTROPIN signaling and flowering under 16 h light:8 h dark (LD16:8, light intensity: ~100 μmol m$^{-2}$s$^{-1}$) at constant temperature, either 20 °C or 15 °C [see spectrum in Supplementary Fig. 1A]. We found that two independent alleles of *PHOT2* (*phot2* = *phot2-101* (SALK_142275), and *phot2 gl1* = *phot2-1 gl1*) showed a moderate but consistent increase in the number of leaves at flowering (Total Leaf Number, TLN) relative to controls [Fig. 1A, B]. While we observed a slight difference at 20 °C, this effect was particularly pronounced at 15 °C. Two alleles of *phot1* in these respective genetic backgrounds did not show consistent differences in leaf number relative to controls [Fig. 1A, B]. We tested other known

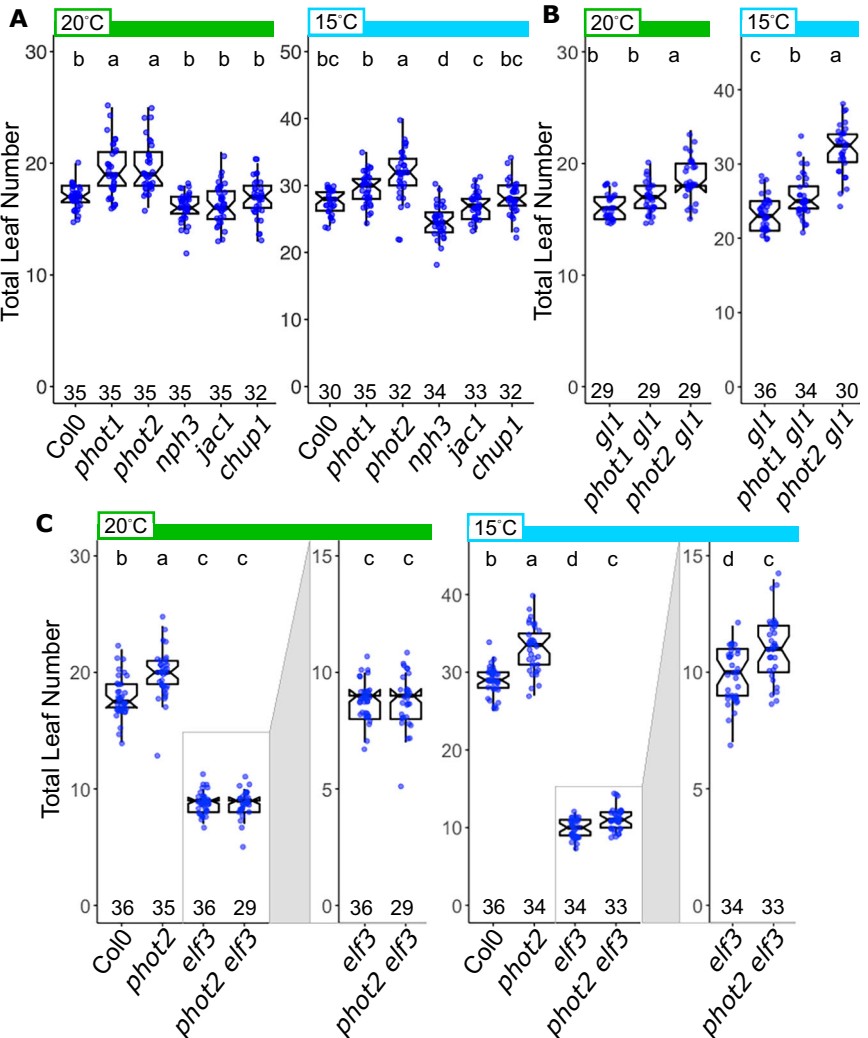

**Fig. 1 | PHOTOTROPIN signaling regulates temperature-labile flowering.**
**A**–**C** Flowering assayed by total leaf number at the time of bud appearance in the indicated genotypes grown in the indicated temperature conditions (20 °C - green bars, 15 °C - light blue bars). Light conditions: 16 light:8 h dark (LD16:8) photoperiod, light intensity: ~100–110 μmol m$^{-2}$ s$^{-1}$ (see Supplemental Fig. 1). Box plots show the median, 25th and 75th percentiles, and whiskers extending to 1.5*inter-quartile range (IQR). Notches approximate the 95% confidence interval of the median. The number of individuals scored for each genotype/condition is indicated along the x-axis. Different letters indicate statistically significant difference between groups (α = 0.05), within each temperature condition by one-way ANOVA + Tukey HSD. **A** Total leaf number at flowering in PHOT-related signaling factors. Alleles: *phot1* (SALK_146058), *phot2* (SALK_142275), *nph3* (SALK_110039), *chup1* (SALK_129128C), *jac1* (WISCDsLOX457-490P9). **B** Total leaf number at flowering in *phot1-5* and *phot2-1* mutants in the Col-0 *gl1* mutant background. **C** Total leaf number at flowering in Col-0, *phot2* (SALK_142275), *elf3*(*SALKseq_043678.1*), and *phot2 elf3*. Y axes for *elf3* and *phot2 elf3* boxplots are expanded for each temperature in boxes to the right of each panel for clarity.

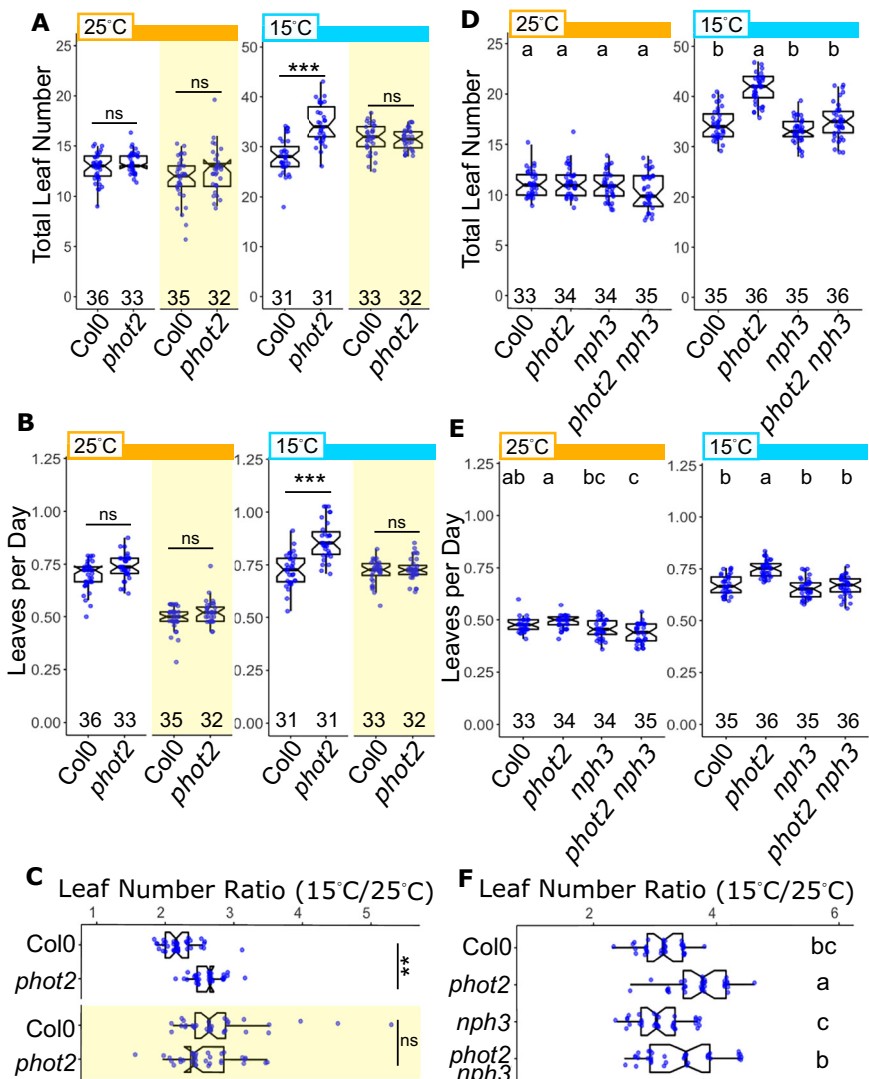

**Fig. 2 | PHOTOTROPIN2 integrates blue light and temperature signals in a light intensity- and NPH3-dependent manner. A–C** Flowering assayed under white light (WL, ~ 100–110 µmol m$^{-2}$ s$^{-1}$ - white background) + yellow filtered light (YL - yellow shaded regions), which specifically reduces the blue component. Graphs in (**A–C**) are derived from the same data set (see N values in panel **A**). The two-way ANOVA + Tukey HSD (genotype X light) result is shown. ns = not significant at $p = 0.05$, *** $p < 0.001$, ** $p < 0.01$. **D–F** Flowering assayed under white light (WL, ~ 50–55 µmol m$^{-2}$ s$^{-1}$ - white background). Graphs in (**D–F**) are derived from the same data set (see N values in panel **D**). Letters indicate statistical groupings by one-way ANOVA + Tukey HSD within the temperature. **A, D** Total leaf number at flowering under indicated light and temperature conditions (25 °C - orange bars, 15 °C -

light blue bars). Alleles: *phot2* (SALK_142275), *nph3* (SALK_110039). See Supplementary Fig. 1 for representative spectra. **B, E** Leaves per Day (Total Leaf Number / Days Post-Germination) calculated from the experiments in (**A**) and (**D**), respectively. **C, F** Leaf number ratio temperature responses calculated from (**A**) and (**D**), respectively. Points do not represent single plants, but a calculated index of the difference in leaf number between two populations. N(15 °C)/N(25 °C) for each genotype and condition: Col-0 WL 31/36; *phot2* WL 31/33; Col-0 YL 33/35; *phot2* YL 32/32. See Source Data. All box plots show the median, 25th and 75th percentiles, and whiskers extending to 1.5*interquartile range (IQR). Notches approximate the 95% confidence interval of the median. The number of individuals scored for each genotype/condition is indicated along the x-axis.

PHOT signaling components, finding that *nph3* showed slightly reduced TLN relative to control [Fig. 1A]. Two mutants with strong effects on chloroplast movement, *chup1* and *jac1*, were similar to controls, indicating that defects in chloroplast movement are unlikely to be responsible for the phenotype [Fig. 1A]. The two phototropins are known to work redundantly or antagonistically, depending on the pathway. The double mutant also displays phenotypes not seen in either single mutant. The *phot1 phot2* double mutant was therefore tested in our flowering assay. We repeated our earlier results, showing no difference between phot1 and Col-0, and increased leaf number in phot2 at 15 °C, but not 25 °C [Supplementary Fig. 2D]. The *phot1 phot2* double mutant, constructed from the SALK insertion lines, showed a TLN phenotype comparable to *phot2* alone, suggesting PHOT2 is the primary actor under these conditions [Supplementary Fig. 2D].

The circadian evening complex component ELF3 is a central player in the floral transition as a repressor of the florigen FT, and is a temperature sensor itself, with the mutant showing reduced sensitivity to cool ambient temperature[32,33]. We therefore asked if the hypersensitivity to low temperature in *phot2* would persist in the absence of ELF3. We found that the flowering times of *elf3* and *phot2 elf3* were identical at 20 °C, both flowering with fewer leaves than Col-0 and *phot2* [Fig. 1C]. However, at 15 °C, *phot2 elf3* flowered with more leaves than *elf3*. Thus, temperature sensitivity unmasked in the *phot2* mutant most likely occurs downstream of, or in parallel with, ELF3 function.

We then tested an expanded set of temperature and light conditions. We examined *phot2* mutants under 25 °C, finding that the mutants were indistinguishable from Col-0 controls, while again observing increased TLN in *phot2* under 15 °C [Fig. 2A, white panels].

We filtered out the blue part of the spectrum (Yellow Filter, YF) and found that Col-0 plants behave identically to *phot2* under 15 °C [Fig. 2A, yellow panels, Supplementary Fig. 1B]. Calculating the rate of leaf generation as Leaves per Day (LPD), we found that both the number of leaves and the rate of leaf generation were identical in Col-0 and *phot2* plants in 15 °C YF vs white light (WL) [Fig. 2B]. Leaf Number Ratio (LNR), the ratio between TLN in 15 °C and 25 °C, was larger in *phot2* than Col-0 in WL, but similar between the genotypes in YF, matching *phot2* in WL [Fig. 2C][34]. Increasing the blue or red components of low WL ($\sim$50 $\mu$mol m$^{-2}$s$^{-1}$) did not modify the high TLN phenotype of *phot2* plants in 15 °C, suggesting that the phenotype derives from the intensity of blue light up to a point of saturation, rather than blue:red ratio or other relative proportions in the spectrum [Supplementary Figs. 1C, D, 2A, B].

Given that PHOT2 is known to work with NPH3, and that *phot2* and *nph3* showed opposing TLN phenotypes [Fig. 1A], we asked if *nph3* might be epistatic to *phot2*. We examined Col-0, *phot2*, *nph3*, and *phot2 nph3* under 25 °C and 15 °C and assayed TLN at flowering (pooled WL(50 $\mu$mol m$^{-2}$s$^{-1}$) data from the experiments in Supplementary Figs. 2A, B). All four genotypes were indistinguishable under 25 °C. Under 15 °C, *phot2* showed increased TLN as expected, *nph3* was similar to Col-0, and *phot2 nph3* showed clear suppression of the *phot2* phenotype [Fig. 2D]. This genetic relationship persisted under 100 $\mu$mol m$^{-2}$s$^{-1}$ WL [Supplementary Fig. 2D]. Interestingly, *nph3* showed its own differential light sensitivity under 15 °C; *nph3* flowered with a similar TLN as Col-0 under low WL (50 $\mu$mol m$^{-2}$s$^{-1}$), but with increasing light intensity (100 $\mu$mol m$^{-2}$s$^{-1}$) it showed reduced TLN relative to control [Supplementary Fig. 2C]. This trend persisted in the experiment in Supplementary Fig. 2D. This suggests that NPH3 may act to counter light-intensity-dependent promotion of flowering, although this light-intensity dependence requires more detailed examination. Together, these data strongly support the hypothesis that PHOT2 acts with NPH3 as part of a light-temperature signaling integration system, using the combination of blue light intensity and low temperature information to buffer or fine-tune development, likely by antagonizing parallel light signals that feed into flowering time control pathways.

## Non-additive genetic interaction between PHOT2 and CAMTA2

We performed an RNA-sequencing experiment using *gl1*, *phot1 gl1*, and *phot2 gl1*, grown under either 20 °C or 15 °C and sampled at lights-off (ZT16) and the following lights-on (ZT24)[Supplementary Data 1]. We identified differentially expressed genes (DEGs), and examined the promoters of these genes using ELEMENT, finding that the promoters of DEGs in *phot2* in 15 C at ZT16 were significantly enriched for variants of CAMTA transcription factor binding motifs (CGCG)[Supplementary Fig. 3A, B][35,36]. Examination of available DNA Affinity Purification (DAP)-seq data revealed enrichment for CAMTA1 and CAMTA5 binding events among these genes [Supplementary Fig. 3C][37]. None of the 75 other TFs with a similar number of reads to CAMTA1 in this database showed similar enrichment, indicating a highly specific effect [Supplementary Fig. 3D].

We screened a panel of *camta* mutants in our flowering assay. We found increased TLN in lines with the *camta2* mutation, with the strongest effect in the *camta1 camta2* double mutant [Fig. 3A]. Mutations in *camta1* and *camta3* did not correlate with increased TLN. Our data thus implicate *CAMTA2*, with possible redundant contribution of *CAMTA1*, as the primary members of this group regulating TLN at flowering. We hypothesized that *PHOT2* and *CAMTA* genes may interact genetically. We observed that *phot2 camta1* plants flower with more leaves than *camta1* alone, an effect similar to *phot2* vs Col-0 control [Fig. 3B]. However, *phot2* failed to increase leaf number in the *camta2* background, with *phot2*, *camta2*, and *phot2 camta2* being statistically indistinguishable under either growth temperature. We also found *phot2 camta1 camta2* TLN to be the same as *camta1 camta2*. We asked if the *camta2* mutant would respond to warmer temperatures in a manner similar to *phot2*. We assayed flowering at 25 °C and 15 °C, observing that all genotypes grown under 25 °C show indistinguishable TLN at flowering, but *phot2*, *camta2*, and *phot2 camta2* show more leaves than the control under 15 °C [Supplementary Fig. 4A]. Together, these data demonstrate that PHOT2 and CAMTA2 regulating the temperature sensitivity of flowering. The non-additive genetic interaction between PHOT2 and CAMTA2 suggests that these factors operate in a coordinated manner.

## PHOT2 and CAMTA2 co-regulate gene expression across the day

We carried out a full day time course RNA-seq analysis, sampling Col-0, *phot2*, *camta2*, and *phot2 camta2* plants every four hours. To ensure close molecular correlates with our flowering assay, we sampled aerial tissue from soil-grown plants[38]. *PHOT2* and *CAMTA2* expression patterns across the day matched the Col-0 control in *camta2* and *phot2* mutants, respectively [Supplementary Fig. 4B], arguing against the hypothesis of transcriptional feedback between these genes.

The temporal occurrence of DEGs identified in *phot2*, *camta2*, and *phot2 camta2* were similar, with a high proportion identified in the mid-day [Fig. 3C]. Examining the DEGs identified in common among all three mutant lines across the day (common DEGs), we observed that genes that were misregulated tended to be identified particularly in the mid-day under both 20 °C and 15 °C, with down-regulation under 15 °C more often observed sightly earlier than up-regulation (ZT8 vs ZT12) [Fig. 3D]. DEGs identified in *phot2*, *camta2*, and *phot2 camta2* plants were highly overlapping within each time point, with evidence of more misregulated genes in the double mutant [Supplementary Fig. 6]. We found a highly significant overlap of genes regulated in the same direction (up or down relative to Col-0 control) in *phot2* and *camta2* in both temperatures [Fig. 3E]. We cannot make a definitive claim as to which genes are directly regulated by CAMTA2. However, examination of the promoters of the genes identified as common DEGs in *phot2*, *camta2*, and *phot2 camta2* under 15 °C using the Motif Finder tool of ELEMENT, 43 genes contain the consensus CAMTA-binding motif (CGCG) and 126 genes contain the CAMTA2 variant motif CGTG, often in multiple instances within 1 kb of the transcription start site [Supplementary Fig. 4C and Supplementary Data 2][35,36,39]. Several genes known to modify flowering, light, or temperature responses in mutant or over-expression lines are present in among the DEGs in the *phot2* and *camta2* mutants, including AGL24 (floral integrator[40]), FGT2 (regulator of heat stress memory[41]), CRF3 (cold-induced AP2-family transcription factor associated with auxin transport[42,43]), ABS2 (RAV-family transcription factor with multiple effects on development, including increased leaf number at bolting upon over-expression[44]), and ARP9 (member of the INO80 chromatin remodeling complex[45]) [Supplementary Data 2]. The relative contributions of these mis-expressed genes to the *phot2* and *camta2* phenotype are unknown. Many others have potential peripheral connections to flowering through hormone signaling, protein interactions, or other relationships, but these remain to be functionally confirmed.

Production and transport of the florigen FLOWERING LOCUS T (FT) is a strong determinant of flowering time in response to environmental conditions[46,47]. The FT family consists of six members, including the floral promoters *FT* and *TWIN SISTER OF FT* (*TSF*), the floral repressors *ARABIDOPSIS THALIANA CENTRORADIALIS* (*ATC*) and *TERMINAL FLOWER 1* (*TFL1*), and the less-well characterized *BROTHER OF FT* (*BFT*) and *MOTHER OF FT* (*MFT*)[48]. All members of this family are lowly expressed, and were not identified as differentially expressed at the FDR < 0.05, FC > 0.25 threshold (with the single exception of *phot2* at 20 C, ZT12) [Supplementary Fig. 5 and Supplementary Data 2]. We observed qualitative differences in *TSF*, *ATC*, and *BFT* expression between Col-0 and our mutant strains, with *FT*, *TFL1*, and *MFT* showing similar expression patterns between genotypes, particularly at 15 °C [Supplementary Fig. 5]. We confirmed *TSF* and *ATC* expression by quantitative Real Time (qRT) PCR, observing trends of reduced *TSF*

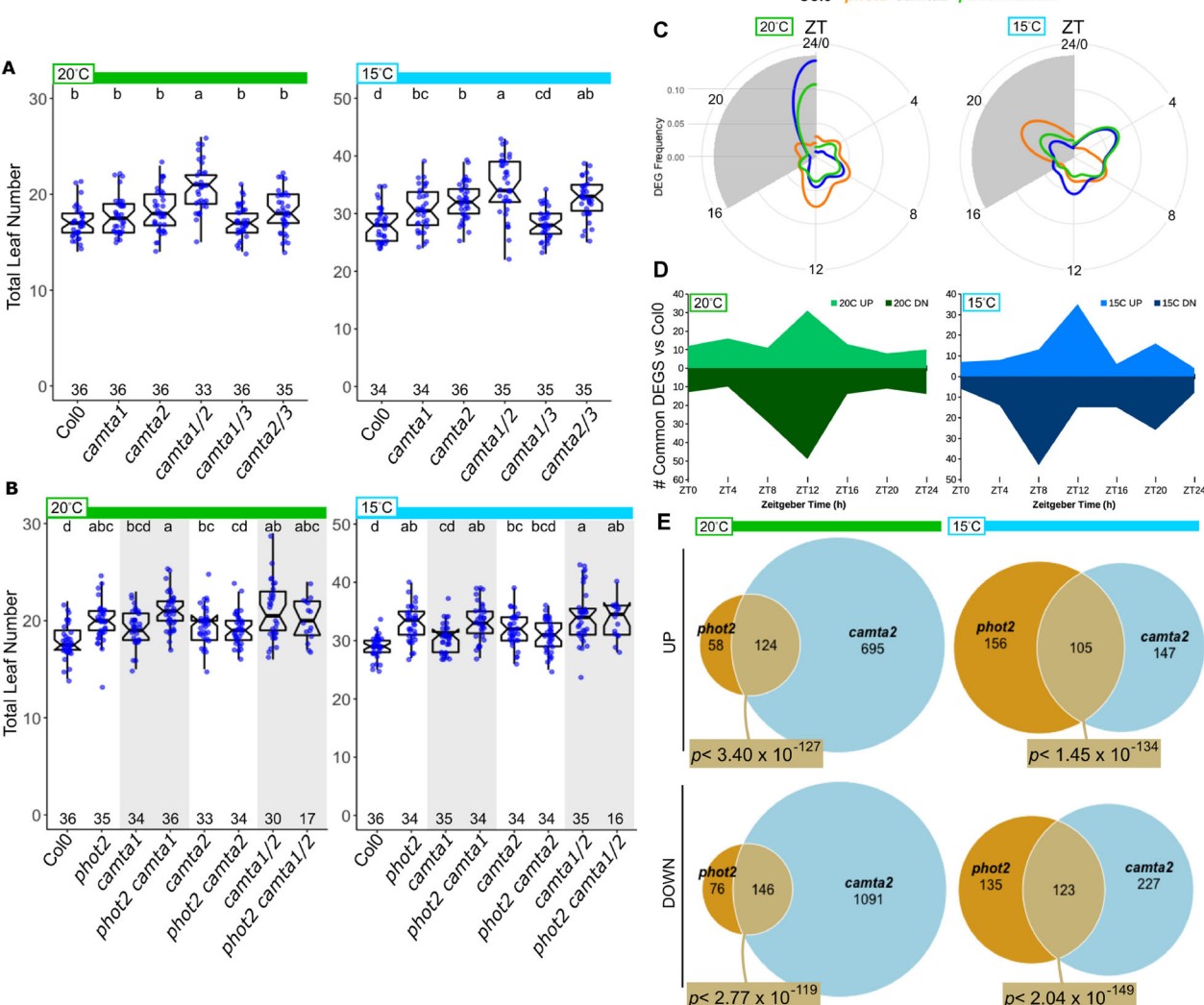

**Fig. 3 | Non-additive genetic interaction between *phot2* and *camta2*. A, B** Total leaf number at flowering under indicated temperature conditions (20 °C - green bars, 15 °C - light blue bars), as in Fig. 1. The *camta2-1* allele is used in (**A–E**). **A** Flowering assayed in *camta1*(SALK_008187), *camta2*(SALK_007027), *and camta3*(SALK_001152) single and multiple mutants. **B** Flowering assayed in *camta1* and *camta2* mutants +/− *phot2* (SALK_142275). Pairs of genotypes in white and shaded areas indicate key comparisons. **C–E** Col-0, *phot2*(SALK_142275), *camta2*(-SALK_007027), and *phot2 camta2* plants were grown on soil at either 20 °C or 15 °C. Samples were collected every four hours, starting at ZT0 (lights-on) of day 10 for RNA sequencing. Differentially expressed (DE) genes were identified between genotypes, within each timepoint, and temperature. Threshold for DE was |0.25| fold-change, FDR < 0.05. **C** The number of genes identified as differentially expressed (DEG Frequency: *y*-axis on radius) in *phot2*, *camta2*, and *phot2 camta2* plants compared to Col-0 control within each timepoint and temperature,

expressed as a circular histogram to emphasize time of day. The *x*-axis (clockwise around the circle) is in Zeitgeber Time (ZT), where ZT0 is the time of lights-on, ZT16 is lights-off, and ZT24 is the next lights-on. The gray region indicates the dark period (ZT16-24). **D** Graphs show the number of up- or down-regulated DE genes identified in common in *phot2*, *camta2*, and *phot2 camta2* plants, plotted by time point and temperature. Lighter color (above 0) indicates up-regulated genes, and the darker color (below 0) indicates down-regulated genes at each time point across the day. Some DE genes were identified as differentially expressed at multiple time points, and are therefore counted in more than one time point (see Supplementary Data 2). **E** Venn diagrams of UP- and DOWN-regulated genes identified in *phot2* and *camta2* mutant plants relative to Col-0. DE gene lists for each genotype across the day were collapsed into a single list and compared between genotypes. *P*-values indicate the significance of overlap by the hypergeometric test (19,139 total genes after expression level filtering).

---

and increased *ATC* in the late afternoon and evening time points (ZT8-16) [Supplementary Fig. 6A, B]. We particularly note that *TSF* and *ATC* are down- and up-regulated, respectively, in a manner consistent with our late flowering phenotype. We calculated the *TSF/ATC* ratio for these samples, finding a consistent reduction of this ratio in our mutants [Supplementary Fig. 6C, D]. We therefore suggest that an altered florigen/anti-florigen balance may contribute to the flowering time delay in *phot2* and *camta2*. Few DEGs identified in our mutants fall into central flowering time determination pathways, with some showing minor differences below our cutoff threshold. We examined a set of these genes, guided by the hypothesis that some differences may be visible in different developmental stages. We examined the floral

integrator *SUPPRESSOR OF CONSTANS 1* (*SOC1*), the thermosensory input gene *SUPPRESSOR OF VEGETATIVE PHASE* (*SVP*), and the floral determination factor *SEPELLATA 3* (*SEP3*) by qRT-PCR on day 15 at ZT8, the shared peak of expression of these transcripts in our day 10 data set. We compared *phot2*, *camta2-1* and *camta2-2* to Col-0 at 25 °C and 15 °C. We noted a slight reduction in SOC1 in the evening-night timepoints at 15 °C on day 10 [Supplementary Fig. 7A]. We observed slightly reduced expression in the mutants at 25 °C and a slight increase in the *camta2* mutants at 15 °C. Expression of SVP showed slight, in some cases statistically significant differences in both temperatures, consistent with the minor differences evident in day 10 [Supplementary Fig. 7B]. SEP3, the most downstream component, showed the most

consistency between days 10 and 15, with trends toward reduced expression in the RNAseq and qRT-PCR under all conditions tested [Supplementary Fig. 7C]. Reduced SEP3 is consistent with a delayed transition from vegetative to floral meristem. It is important to note that the whole-shoot samples used here for transcript analysis may be subject to dilution effects when examining genes that are operating in the meristem. Future studies will need to examine expression at a more detailed tissue or cell level.

We reasoned that system-level regulation may be more visible when analyzed across the complete time series, rather than single time-point comparisons. We identified co-expressed "communities" of genes in our data set [Fig. 4A and Supplementary Data 3, "Methods"]. Mean scaled expression levels of each gene in the resulting communities were averaged, and communities 3, 6, and 15 are shown [Fig. 4B]. We compared the scaled expression in *phot2*, *camta2*, and *phot2 camta2* to Col-0 at each time point to identify communities that were systematically mis-regulated, finding time- and temperature-dependent differences [Fig. 4C]. Gene Ontology analysis of these communities reveals system-level regulation of specific processes by PHOT2 and CAMTA2, and provides clues as to potential pathways that may control TLN. Community 3 shows global up-regulation of ribosome biogenesis-related processes [Fig. 4D]. This pattern persists when limiting the gene set to the GO category Ribosome Biogenesis [Fig. 4E]. This community is more strongly up-regulated in *camta2* and *phot2 camta2* than in *phot2*. Community 6 shows up-regulation of cell cycle-related processes across the day in the mutants, and the pattern is maintained, limiting the gene set to GO: Cell Cycle [Fig. 4F, G]. Community 15 shows up-regulation of DNA replication-related processes in the mutants, most clearly in the evening and nighttime points [Fig. 4H]. While there is only one statistically different time-point seen in Community 15 at 15 °C as identified by the network [Fig. 4C], refining it to the members of the GO category DNA Replication reveals clear up-regulation in the mutants [Fig. 4I]. These trends are preserved in the Ribosome Biogenesis, Cell Cycle, and DNA Replication groups when examined at the single gene level [Supplementary Fig. 8]. Literature searches did not reveal known components involved in the regulation of flowering time or instances of genes in these communities being assayed for flowering time. This unbiased network building and analysis strategy allows us to resolve system-level effects on fundamental growth-related modules by PHOT2 and CAMTA2 [Fig. 4E, G, I] and suggests a possible method of controlling leaf generation rate in parallel with classical flowering time control.

Two additional communities show significant differences between wild-type and mutant time courses: Community 1 is comprised of high-amplitude cycling transcripts with peak expression at ZT0/24 and a trough at mid-day. Community 1 is enriched for GO categories related to light signaling, and also includes central members of the circadian clock. Differential expression in the mutant plants was identified at ZT4-8 at 15 °C, as well as around ZT0-24 at 20 °C [Fig. 4C and Supplementary Fig. 9]. While the magnitude of this change is small and occurs at a low point of expression, a perturbation in the circadian clock or associated factors is certainly of interest, as it has a strong influence on flowering time. Community 11 shows a strong low-temperature-induced peak at ZT16 that is enriched for genes involved in flavonoid biosynthesis [Supplementary Fig. 9]. This peak is further enhanced in the *phot2* and *camta2* mutants. Connections between flavonoid production and flowering are unclear, although flavonoids are suspected to protect against oxidative stress and FT-family members may influence both processes[49]. Flavonoid mutants can effect both root and shoot architecture, as well as cold tolerance, suggesting broad roles for flavonoids in environmentally regulated growth[50,51].

In sum, the gene expression patterns observed in *phot2* and *camta2* mutants reveal mild but broadly distributed effects. Those genes that are known to be directly involved in the control of flowering time and identified as differentially expressed show relatively minor differences. This distribution of targets suggests systemic modulation of growth through multiple pathways.

## EHB1: Potential CAMTA2- and temperature-dependent input to the PHOT signaling system

Light input to the PHOT/CAMTA/NPH3 system most likely comes directly through PHOT photosensors, but temperature input is known to effect many parallel processes. Given the links that we have seen between PHOT2 and CAMTA2, and the known functions of CAMTA genes in low-temperature-dependent gene expression, we searched our RNAseq data set for temperature-dependent gene expression patterns that were disrupted in the *camta2* mutant. We found that expression of the calcium-dependent lipid binding protein *ENHANCED BENDING* 1 (*EHB1*) was up-regulated at lower temperature in Col-0, with reduced peak expression in *camta2* [Fig. 5A]. Expression in *phot2* was similar to Col-0 control under both temperatures. Quantitative RT-PCR in an independent RNA null allele of *CAMTA2* (*camta2-2*) shows that low temperature-dependent induction of *EHB1* transcript is largely dependent on CAMTA2 [Fig. 5B]. We tested if CAMTA2 could bind to the EHB1 promoter, finding that an N-terminal fragment of CAMTA2, containing the DNA-binding domain, binds to a 23 bp fragment of the proximal EHB1 promoter in vitro. This fragment contains the canonical CAMTA-binding motif (CGCG). Wild-type unlabeled competitor DNA strongly reduces signal, and mutant (CGCG > ATAT) competitor DNA can partially reduce signal in 200x molar excess. CAMTA2 is unable to bind a mutated probe [Fig. 5C]. This suggests that the core four-base motif is not the only feature recognized by CAMTA2, or a CAMTA2-containing complex. Determination of CAMTA2 binding properties at the EHB1 promoter in vivo, and temperature-dependent chromatin occupancy in general, are important areas of future study. EHB1 acts as a negative regulator of NPH3-dependent phototropic and gravitropic growth via direct interaction with the BTB domain[52]. This, combined with our expression data in the *camta2* mutant, led us to hypothesize that plants lacking EHB1 would show increased TLN under low temperature. We tested two alleles of *camta2* and two alleles of *ehb1* under 25 °C and 15 °C. These mutants were similar to Col-0 controls under 25 °C, but show comparable increases in TLN under 15 °C [Fig. 5D].

EHB1 and NPH3 have been described as interacting at the phenotypic and biochemical levels[52]. We examined co-localization of EHB1 and NPH3 using transient expression in tobacco leaves. In dark-adapted leaves, YFP-NPH3 shows small puncta and diffuse membrane-adjacent localization, and EHB1-mCherry shows diffuse membrane or cytoplasmic localization [Supplementary Fig. 10A, Dark]. Single confocal sections show overlapping signals, suggesting possible co-localization near the membrane [Supplementary Fig. 10B, Dark]. After repeated scans with the 488 nm (blue) laser, which activates PHOT signaling, NPH3 condenses into large puncta. There are not obvious changes in EHB1 localization, nor is there substantial overlap with the NPH3 condensates [Supplementary Fig. 10A, B, scan 10]. While this experiment has limited resolution, it does suggest that EHB1 may exert its influence on NPH3 during the membrane-association phase, rather than the cytosolic condensate phase of the signaling cycle. Together, these data are consistent with a role for EHB1 in transducing temperature information from CAMTA2 to the PHOT pathway, where it modulates NPH3 activity [Fig. 6].

## Discussion

Plants set and regulate temperature responses to influence developmentally and agronomically relevant traits. In this study, we demonstrate light and temperature-dependent control of flowering time by the PHOT2 blue photoreceptor. PHOTOTROPINS have been described as having no role in flowering since their identification[8,18]. Typical laboratory growth temperatures for *Arabidopsis* lie in the 20–23 °C range. We observed increased TLN in *phot2* mutants at 15 °C,

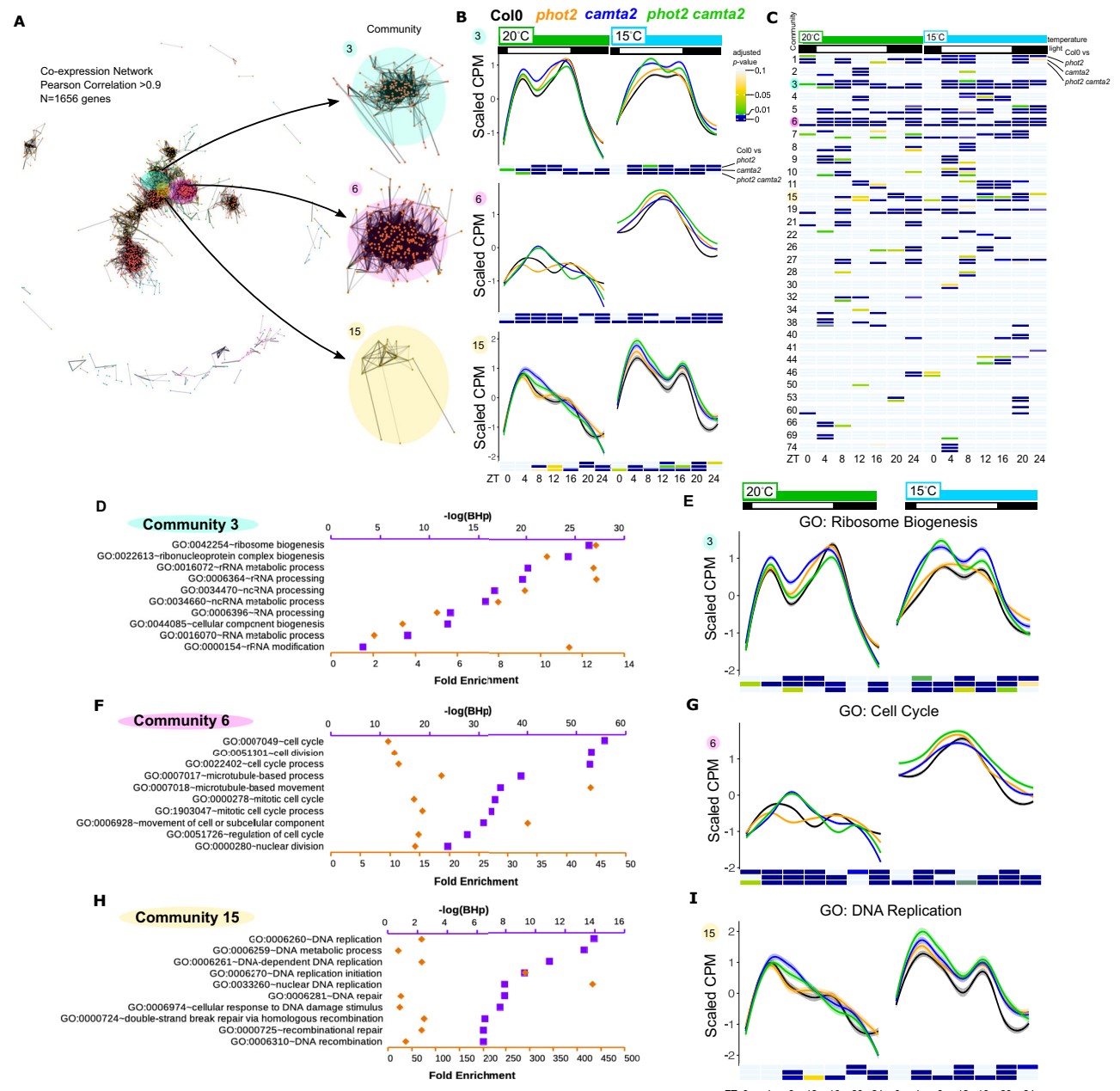

**Fig. 4 | System-level control of basic growth processes by PHOT2 and CAMTA2.**
**A** Coexpression network constructed from genes with a Pearson Correlation Coefficient > 0.9 (N = 1656 genes). Communities 3, 6, and 15 of the 81 identified communities are magnified and shown at right. **B** Mean scaled timecourse expression patterns for communities 3, 6, and 15 are shown by time of day, genotype, and temperature. Traces are colored by genotype. Col-0: black, *phot2* (SALK_142275): orange, *camta2*(SALK_007027): blue, *phot2 camta2*: green. The shaded area around the traces are 95% confidence interval. Black and white bars at the top indicate the light-dark cycle. Temperature is shown in green and light blue bars at the top. Heatmaps below each graph are *p*-values derived from 2-way ANOVA+Tukey HSD test (genotype X time of day within temperature) for each mutant compared to Col-0 and are also included in panel (**C**). Each heatmap has three rows: the first is *phot2* vs Col-0, the second is *camta2* vs Col-0, and the third is *phot2 camta2* vs Col-0 (labeled at right of Community 3). Color key for heatmap is at left, and is the reference for panels (**B**, **C**, **E**, **G**, and **I**). **C** Heatmap of

*p*-values derived from 2-way ANOVA + Tukey's HSD test (genotype X time of day within temperature) for each mutant compared to Col-0 for all identified co-expression communities made up of at least four genes. The heatmap is divided by temperature (green and blue bars at top) and by community (numbers at left). Color bar at left shows color mapped to p-value. Communities 3, 6, and 15 are highlighted for comparison to (**B**). **D**, **F**, **H** Gene ontology for communities 3, 6, and 15, respectively. The top ten most significant GO-Biological Process terms for each community are shown, with -log(Benjamini-Hochberg *p*-value) [-log(BHp)] mapped to the upper axis in purple, and Fold Enrichment mapped to the lower axis in orange. Significance of enrichment was determined by Fisher's Exact Test and corrected for multiple comparisons (Benjamini-Hochberg) in DAVID. **E**, **G**, **I** The means of all samples within the top GO Biological Process category from each community are plotted as in (**B**). **E** Community 3, Ribosome Biogenesis. **G** Community 6, Cell Cycle. **I** Community 15, DNA Replication.

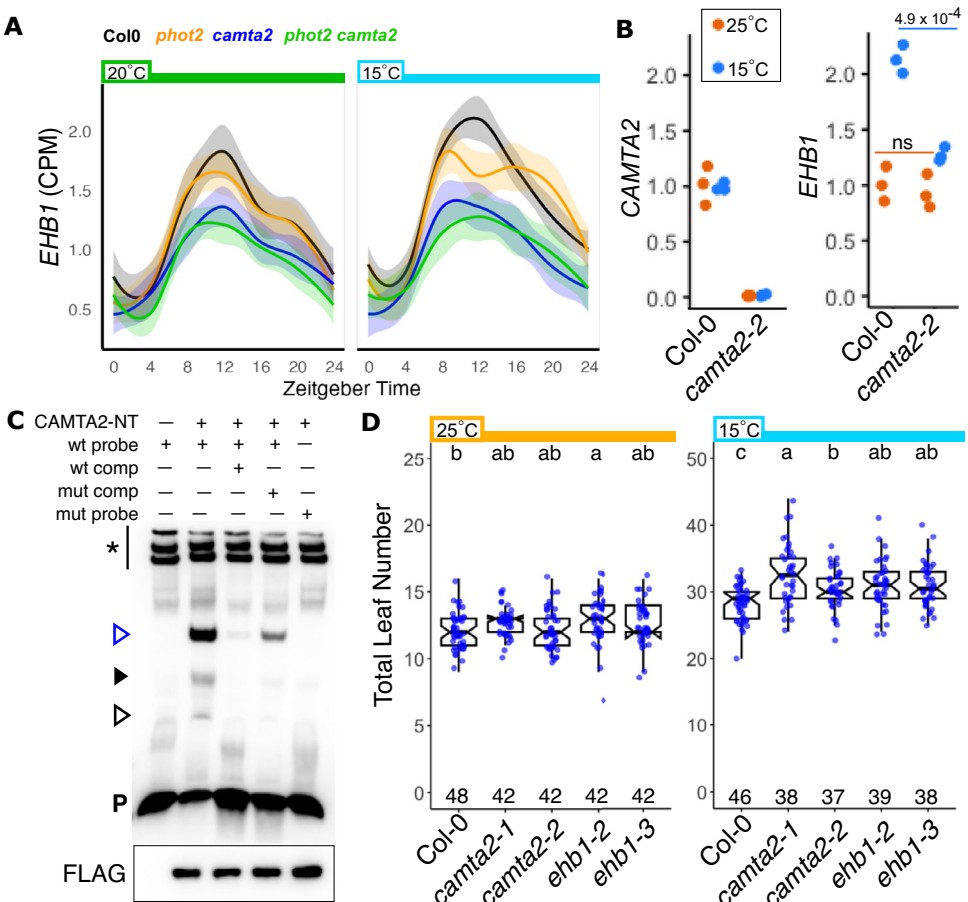

**Fig. 5 | EHB1 connects CAMTA2 temperature input to PHOT signaling. A** EHB1 transcript time courses plotted as Counts Per Million reads (CPM) for plants grown at 20 °C (green bar) and 15 °C (light blue bar). Traces (mean CPM) are colored by genotype. Col-0: black, *phot2*: orange, *camta2*: blue, *phot2 camta2*: green. Shaded areas around traces are the 95% confidence interval. **B** Taqman probe-based qRT-PCR of CAMTA2 and EHB1 in Col-0 and *camta2-2* (SALK_139582) under 25 °C (orange dots) and 15 °C (blue dots). ns = not significant and *** = $p < 0.001$ by two-tailed Student's *T* Test. **C** CAMTA2-NT (N-terminus - aa1-364) was expressed in wheat germ extract for Electrophoretic Mobility Shift Assay (EMSA). A 23 bp fragment of the EHB1 promoter containing the core CAMTA-binding motif (CGCG) was used as the target sequence. Mutant competitor and probe changed CGCG to ATAT. Unlabeled competitors were supplied at 200x molar excess over the labeled probe. The first lane contains unprogrammed wheat germ extract (WGE). * = endogenous biotiny-lated proteins in WGE; Open blue arrowhead = likely primary complex containing CAMTA2; Solid black arrowhead = light band is non-specific from WGE, with over-lapping possible CAMTA2 and wt promoter-dependent complex; Open black arrowhead = possible CAMTA2 monomeric binding event. P = free probe. Done twice with similar results. The lower panel is the CAMTA2-NT-FLAG protein in the EMSA reactions. **D** Flowering time assay of *camta2* (1:SALK_007027, 2:SALK_139582) and *ehb1* (2:SAIL_385_C07, 3:SAIL_1307_C10.v1) mutants. Flowering was assayed and boxplots presented as in Fig. 1. Letters indicate statistical groupings by one-way ANOVA between genotypes, within temperature conditions.

a condition under which *Arabidopsis* grows quite slowly and is rarely tested. This may explain why a role for PHOT2 in flowering regulation was overlooked. Our modifications of environmental light conditions under warm and cool temperatures demonstrate PHOT2- and NPH3-dependent light-temperature signal integration. Given that NPH3 gates the influence of PHOT2 [Fig. 2D–F], and the *nph3* early flowering phenotype is light intensity-dependent [Supplementary Fig. 2C], we hypothesize that NPH3 acts to inhibit other light-dependent signals that act to promote flowering, and that PHOT2 negatively regulates this function of NPH3 [Fig. 5E].

Thermal and spectral sensitivity of flowering is widely distributed among light signaling pathways. Blue light acts through seven of the 13 non-photosynthetic photoreceptor proteins in *Arabidopsis*. FLAVIN-BINDING KELCH REPEAT F-BOX 1 (FKF1) controls flowering by degrading a transcriptional repressor of CONSTANS (CO)[53–55]. ZEITLUPE (ZTL) and LOV KELCH PROTEIN 2 (LKP2) regulate flowering under short days[56]. Recent work has described blue light via CRYPTOCHROME 2 (CRY2) feeding into temperature-regulated flowering by controlling splicing of flowering regulators in a day-length-dependent manner[57]. Plants lacking red/far-red sensing

PHYTOCHROME (PHY) B and/or D proteins flower with fewer leaves than controls in short days under 22 °C, but are similar to controls under 16 °C[58,59]. PHYB also feeds into a shade-high temperature growth regulation pathway, suggesting multiple light-temperature coincidence detectors operate across different processes[60]. Our data suggest that the PHOT2-NPH3 system may interact with one or more of these photoreceptor signaling pathways, cross-regulating incoming light and temperature information, and/or buffering against fluctuations. Given that reducing the blue component of white light eliminates the difference in TLN between *phot2* and Col-0 [Fig. 2A–C], but increasing blue or red failed to modify the phenotype [Supplementary Fig. 2A, B], we prefer the hypothesis that PHOT2-NPH3 signaling acts to gate the influence of low-intensity blue light signaling pathways. Consistent with this, recent work has indicated that *Arabidopsis* senses gradual light intensity changes at dawn and dusk[61]. This work, therefore, provides a unique position from which to map out the processing and integration of light and temperature information between photoreception systems.

We characterized flowering time in a suite of mutants covering PHOT-interacting genes, finding a role for NPH3. We show that NPH3 is

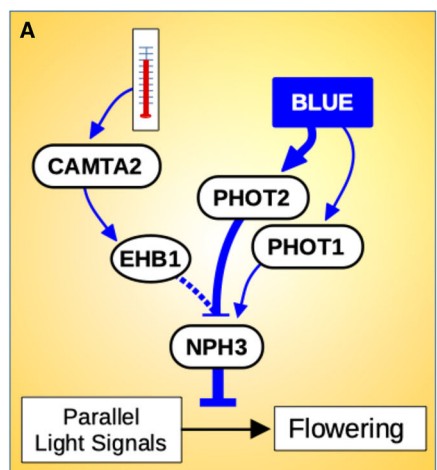
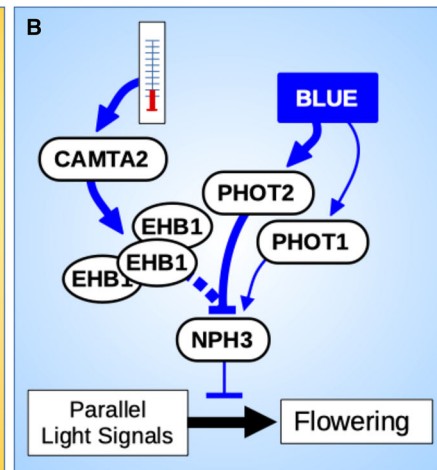

**Fig. 6 | A hypothetical model of the PHOT2-CAMTA2 light-temperature coincidence detector.** A model of the PHOT-CAMTA2-NPH3 light-temperature coincidence detector under warm (**A** -orange) and cool (**B** - blue) temperatures. PHOT1 promotes and PHOT2 inhibits the active form of NPH3. NPH3 negatively regulates light signals that promote flowering. CAMTA2-dependent EHB1 accumulation in low temperature inhibits NPH3's ability to oppose pro-flowering signals. In *phot2* mutants, the active form of NPH3 dominates, strengthening its ability to repress pro-flowering signals. In *camta2* or *ehb1* mutants, EHB1 fails to accumulate, allowing NPH3 to more strongly inhibit pro-flowering signals, thus increasing TLN.

epistatic to PHOT2 in flowering regulation[Figs. 1A, 2D–F and Supplementary Fig. 2C]. The NRL family has 33 members in *Arabidopsis*, only a few of which have been characterized experimentally[22]. Upon blue light activation, PHOT1 both directly phosphorylates and triggers dephosphorylation of NPH3, releasing it from the plasma membrane into cytoplasmic complexes[62–66]. The precise function of NPH3 in these cytoplasmic complexes is unknown. PHOT2 and RPT2 subsequently facilitate NPH3's return to the plasma membrane via unknown mechanisms[64,67]. NPH3 and NCH1 form complexes with CULLIN3 to ubiquitinate target proteins, but whether RPT2 forms similar complexes is unclear[26,68]. Few target or client proteins for NRL proteins have been described, and identification of these interactors will be of distinct interest for understanding cross-regulation of signaling pathways by NRL proteins. RPT2 and NCH1 have redundant functions in the regulation of chloroplast movement via JAC1[69,70]. We observe no notable flowering phenotype in plants lacking JAC1 or CHUP1, both of which are critical for chloroplast movement [Fig. 1A]. This indicates that chloroplast movement is highly unlikely to be a central factor in the flowering phenotype.

Multiple links support the hypothesis of signaling between PHOTs, NRL proteins, calcium, and CAMTA TFs. CAMTAs are regulated by intracellular $Ca^{2+}$ and CALMODULIN (CAM) binding[71]. Cold acclimation, in which pre-treatment with low temperature increases survival when exposed to freezing temperature, requires $Ca^{2+}$ mobilization, with application of $Ca^{2+}$ channel blockers or $Ca^{2+}$ chelators resulting in reduced survival in low temperature[72]. The *camta1 camta3* double mutant also shows reduced cold acclimation[24]. A recent study reported NPH3 co-purifying with CAM7, although this interaction was not further examined[62]. A link between PHOT signaling and the CAMTA transcription factors was proposed when NCH1 was identified as a CAMTA3-interacting protein, and the *nch1* mutant was shown to modify CAMTA3-dependent responses to bacterial infection, blocking CAMTA3 ubiquitination and degradation[26]. While a subsequent study in protoplasts failed to corroborate this finding, it is possible that this interaction is dependent on cell-specific conditions or factors[30]. However, studies in potato characterized an NRL protein as a susceptibility factor for *Phytophthora infestans* infection, and showed that blue light and PHOT1 decreased the immune response[73,74]. Our findings are in line with the emerging literature suggesting a series of pathways linking PHOTs, NRL proteins, $Ca^{2+}$, and CAMTA transcription factors to regulate development and immunity.

We identify EHB1, a $Ca^{2+}$-dependent lipid-binding protein, as a temperature-dependent link between CAMTA2 and the PHOT-NPH3 blue light signaling pathway. Low-temperature-dependent *EHB1* transcription in the mid-day depends on CAMTA2, which can bind directly to the *EHB1* promoter [Fig. 5A–C]. EHB1 was described as a negative regulator of NPH3, possibly via direct interaction with NPH3's BTB domain[52]. EHB1 membrane association is enhanced by calcium[75]. Blue light-activated $Ca^{2+}$ transients rely heavily on PHOTs, with PHOT2 likely to be the dominant activator[76–78]. PHOT2-dependent calcium currents may therefore act on EHB1 to promote NPH3 translocation from cytosol to plasma membrane. The observation that *phot2*, *camta2*, and *phot2 camta2* have identical temperature-responsive flowering phenotypes suggests that both factors are necessary for this control mechanism [Fig. 3B and Supplementary Fig. 4]. Plants lacking EHB1 or CAMTA2 have identical low-temperature-dependent flowering phenotypes as well, consistent with the hypothesis that EHB1 is the primary factor linking CAMTA2 to PHOT2 [Fig. 5D]. While we observe CAMTA2-dependent changes in the *EHB1* transcript, the response at the protein level is unknown. CAMTA TFs are also regulated by calcium through CaM binding to the C-terminal domains. It is possible that PHOT2-dependent $Ca^{2+}$ currents can simultaneously enhance EHB1 transcription via CaM-CAMTA2 and membrane localization via $Ca^{2+}$ and EHB1 effects on NPH3, perhaps through membrane binding. Consistent with this, we observe possible co-localization of EHB1 and NPH3 near the membrane in the dark, but not in NPH3 condensates after blue light treatment [Supplementary Fig. 10]. However, the biochemical relationship between EHB1 and NPH3 remains unclear. In addition, analysis of the *ehb1 nph3* double mutant, in particular, may deliver valuable insight into the relationship between these two factors. The dynamic relationships among PHOT2, calcium, CAMTA2, EHB1, and NPH3 will be of definite interest for future study.

Studies of PHOT-dependent gene expression have thus far been restricted to examining the effects of acute blue light treatment[79]. The CAMTA family's effect on transcription is best understood in the context of acute treatment with noxious cold or other stresses[24,35]. Our time-course transcriptomes allow comparisons across the day, and within a relatively narrow ambient temperature band (20 °C vs 15 °C), rather than noxious high or low temperatures. There are six CAMTA TFs in *Arabidopsis*, and CAMTA genes are known to be partially redundant[29,38]. We observe more DEGs in the double mutant than in either single mutant. It is possible that PHOT2 works with more than one CAMTA TF, and that

removing CAMTA2 uncovers more extensive regulation through other family members that is further compromised by the removal of PHOT2. We also observe DEGs in *phot2* that are not regulated by *camta2*, and vice versa, indicating multiple pathways by which these components influence gene expression (Fig. 3E). Our network-based analysis reveals system-level PHOT2- and CAMTA2-dependent regulation of translation, the cell cycle, and DNA replication, among other processes [Fig. 4C–G]. The *phot2* mutant exhibits not only increased TLN but an increase in leaves generated per day until flowering, consistent with increased organ formation, cell division, and growth [Fig. 2C]. Three types of growth are represented here: translation, which increases cytoplasmic volume and cell size, cell division increases cell number, and DNA replication contributes to both cell division and endoreduplication[80,81]. We suggest that the up-regulation of these genetic modules may reflect increased cell division or growth in the *phot2* and *camta2* mutants at low temperature, either through more cells dividing or more rapid division. At whole-shoot resolution, we cannot say how these types of growth are distributed through the plant, and how they are altered in the mutant lines. It will be of considerable interest to examine the division rate, DNA content, and cell size in growing tissues in *phot2* and *camta2* mutants.

Systematic changes in environmental light and temperature across the year provide information that plants decode to fine-tune life history traits. Here, we describe a role for PHOT2, CAMTA2, and NPH3 in the coordinated processing of blue light and low temperature information, feeding into flowering time control. This coincidence detector is positioned as a negative regulator of other flowering-promoting signals in low temperature, providing new insight into environmental signal integration. Examination of this system in a variety of plant species will be of interest, as it may provide an access point for engineering tolerance to a range of temperatures and understanding mechanisms of resilience.

## Methods

### Plant Material

*Arabidopsis* plants are in the Col-0 background unless otherwise noted. *phot1-5 gl1* and *phot2-1 gl1* lines were isolated in the *gl1* mutant background and were described[21,82]. *phot1* (SALK_146058), *phot2* (SALK_142275), *nph3* (SALK_110039), *chup1* (SALK_129128C), *jac1* (WISCDsLOX457-490P9), *elf3* (SALKseq_043678.1), *camta1* (SALK_008187), *camta2* (1:SALK_007027, 2:SALK_139582), *camta3* (SALK_001152), *ehb1* (2:SAIL_385_C07, 3:SAIL_1307_C10.v1) mutants were obtained from the Arabidopsis Biological Resource Center (Ohio State University). Mutant combinations were generated by standard genetic crosses and confirmed by PCR genotyping using primers in Supplementary Data 4.

### Flowering assays

Seeds were surface sterilized using the chlorine gas method in a chemical fume hood for ~1 h, then plated on 1/2x LS + 0.8% phytoagar medium (Caisson) and stratified in the dark at 4 C for 6-7 days. Seeds were germinated in a growth room set on a long-day (LD16:8) photocycle (light intensity ~ 80 μmol m$^{-2}$s$^{-1}$) at ~ 22 °C. Seedlings were transplanted to soil with fertilizer and fungal inhibitor on day three (two days after germination) and moved to growth chambers (Percival) under the indicated growth conditions. Light for flowering assays was provided by cool white fluorescent bulbs (Philips) set at the indicated intensities. Light intensity and spectra were confirmed using the LI-250A light meter or the LI-180 spectrometer (LiCor). Supplemental blue and red light were provided by LED bulbs in addition to the fluorescents. Blue light was specifically reduced by filtering the white fluorescent light with #101 Yellow filter sheeting (Lee Filters). Chamber ambient temperatures were confirmed with a glycerol thermometer (Fisher Scientific) or HOBO temperature loggers. Trays were rotated and cycled around the chamber every 1-2 days to avoid position effects. Flowering time was scored at the appearance of buds at the center of the rosette. Leaves were counted under a dissecting microscope (Leica).

### RNA isolation and RNA sequencing

RNAseq1: Plants were sterilized, stratified, germinated, and transplanted as for the flowering assays. Above-ground tissue from three seedlings per sample were collected and snap-frozen in liquid nitrogen in biological duplicates at ZT16 and ZT24 of day 14 post-germination. Total RNA was extracted with the Qiagen RNeasy Plant mini kit according to the manufacturer's instructions. Single-end sequencing libraries were made and sequenced on the HiSeq2500 at the Salk Institute Next Generation Sequencing core.

RNAseq2: Plants were sterilized, stratified, germinated, and transplanted as for the flowering assays. Above-ground tissue from six seedlings per sample were collected and snap-frozen in liquid nitrogen in biological triplicate at ZT0, 4, 8, 12, 16, 20, and 24 of day 10 post-germination. Total RNA was extracted with the RNeasy Plant mini kit (Qiagen) and DNAse treated (Qiagen) according to the manufacturer's instructions. RNA was poly-A selected, and paired-end libraries were prepared at the Salk Institute Next Generation Sequencing facility. Sequencing data were generated on a NovaSeq 6000 S4-XP with PE100 reads at the University of California, San Diego sequencing core. Samples were demultiplexed using the bcl2fastq Conversion Software (Illumina, San Diego, CA).

### RNA-seq analysis

RNAseq1: Reads were aligned to the TAIR10 *Arabidopsis* genome using TopHat, and reads per gene quantified using HTSeq[83]. Differential expression analysis was carried out in edgeR, with comparisons set between mutant and wild type samples within each time point and temperature condition at False Discovery Rate (FDR) ≤ 0.05 and Fold Change (FC) = | 0.25 |[84].

RNAseq2: Reads were aligned to the TAIR10 *Arabidopsis* genome using STAR with default parameters except --alignIntronMax 2000 and --outFilterMismatchNmax 2. Reads per gene were calculated using the Araport11 list of genes and transposons in HTSeq with default parameters except "-m intersection-strict -s reverse -r pos". Differential expression analysis between mutant and wild-type samples at each time point was carried out in edgeR with False Discovery Rate (FDR) ≤ 0.05 and Fold Change (FC) = | 0.25 |. Genes with cumulative CPM > 1 across all samples and at least one sample with CPM > 1 (defined by "rowSums(cpm(TotalRawCounts_Col0.dge) > 1) ≥1") were retained for analysis. Sample 25 (Col-0, 15 C, ZT12, rep1) was omitted from the final analysis due to evidence of pathogen infection.

### Network co-expression analysis

Network co-expression analysis was carried out in R as described, starting from the complete list of all genes identified to be differentially expressed in any of the mutant lines at any time-point relative to Col-0 (*N* = 3478 genes - see Supplementary Data 3[85]). To more directly compare gene expression dynamics across the day, CPM values were scaled such that the mean expression across all samples for each gene is equal to 0. Pearson Correlation Coefficients (PCC) were calculated between each gene across all samples using the "cor" function (R). Genes that were co-expressed with PCC > 0.9 were used to construct a network using the igraph R package[86]. Co-expressed communities within this network were identified using "cluster_edge_betweenness" in igraph. Gene Ontology enrichment analysis by community membership was carried out in DAVID with an inclusion threshold of Benjamini-Hochberg *p*-value < 0.05[87,88].

### qRT-PCR

RNA was extracted from six 10-day-old, or three 15-day-old, soil-grown seedlings with the RNeasy Mini kit (Qiagen). First strand

synthesis was carried out with 2 μg (day 10) or 3 μg (day 15) total RNA using the Maxima First Strand cDNA Synthesis kit (Thermo) according to the manufacturer's instructions, and 100 ng cDNA equivalent was used per reaction. Real-Time PCR was done using TaqMan Gene Expression Assays (Thermo - Supplementary Data 4) and TaqMan Gene Expression Master Mix according to the manufacturer's instructions. Reactions were carried out in a BioRad CFX Opus 384 Real-Time PCR Machine. Relative transcript abundance was determined by the $\Delta\Delta Cq$ method using the static AT4G26410 transcript as the internal control.

### Electrophoretic mobility shift assays

A plasmid encoding CAMTA2-NT (amino acids 1-364) driven by the SP6 promoter was synthesized (GenScript), and 4 μg of plasmid expressed in the TnT Wheat Germ Extract Kit (Thermo) for 2.5 h at 25 °C. Two μL of programmed or unprogrammed extract was used per reaction. A 23 bp fragment of the EHB1 promoter−biotin labeled probe and unlabeled competitor, wild-type and CGCG > ATAT CAMTA-box mutant−were synthesized (IDT - Supplementary Data 4). Binding was carried out using the LightShift Chemiluminescent EMSA Kit (Thermo) according to manufacturer instructions, with the addition of 2.5% glycerol, 5 mM $MgCl_2$, and 3 μg double-stranded polyAT DNA, with 20 fmol labeled probe and 4 pmol (200x) unlabeled competitor per reaction as indicated. Detection was carried out with the Chemiluminescent Nucleic Acid Detection Module (Thermo) according to manufacturer instructions and imaged on a Sapphire Biomolecular Imager (Azure Biosystems).

### Western blots

Samples from the EMSA reactions were mixed with 2xLDS buffer (Life Technologies) and heated for 5 min at 85 °C. Samples were run on precast 4–12% gel (Life Technologies) and transferred to nitrocellulose membrane. Membranes were blocked with TBST (tris-buffered saline + 0.2% Tween-20) with 3% milk, and incubated with monoclonal anti-FLAG-HRP clone M2 (Sigma, #A8592) at 1:2000 dilution in blocking solution for 1 h shaking at room temperature. membrane was washed 5 x in TBST, then 2 x in water. Detection was carried out with SuperSignal West Pico PLUS (Pierce) according to manufacturer instructions. Membrane was imaged on a Sapphire Biomolecular Imager (Azure Biosystems).

### Tobacco leaf infiltration and confocal imaging

YFP-NPH3 and EHB1-mCherry were cloned into pCHF3[89] using Gibson Assembly (primers in Supplementary Data 4). Transformed agrobacterium GV3101 was grown in liquid culture, collected, and resuspended in infiltration medium (10 mM MES pH 5.6, 10 mM $MgSO_4$, 200 μM acetosyringone) to a final OD of 1.0. *Nicotiana Benthamiana* plants were grown in a greenhouse for 4-6 weeks. Leaves were infiltrated and allowed to express for ~ 48 h. Plants were moved to a dark chamber ~16 h prior to imaging. Leaf disks were collected and mounted on slides under dim green safe light. Confocal imaging was carried out on a Leica SP8 Stellaris. Imaging settings were kept consistent between samples. YFP and/or mCherry positive regions on the leaf disc were identified and imaged as quickly as possible to avoid excess light exposure. Ten scans with the 488 nm laser line were carried out to activate phototropin signaling.

### Statistics

Sample numbers are indicated within each figure. Comparisons between two groups were done with *t* tests (two-sided), and comparisons among three or more groups were done with one-way ANOVA and Tukey's HSD test at alpha = 0.05. Genome X Environment interactions were tested by two-way ANOVA and Tukey's HSD test at alpha = 0.05. Leaf Number Ratio (LNR) was calculated as described[34]. Significant overlap between populations, including binding site enrichment and comparison of DEG lists between genotypes, was determined using the hypergeometric test dhyper implemented in R.

### Reporting summary

Further information on research design is available in the Nature Portfolio Reporting Summary linked to this article.

## Data availability

The data in this paper are included in the figures, Supplementary Data, and Source Data files provided with the paper. RNA-seq data have been deposited at NCBI-GEO under accession code GSE240124. Source data are provided in this paper.

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

## Acknowledgements

We thank Dr. John Christie, Dr. Claudia Oecking, Dr. Christian Fankhauser, and Dr. Michael Thomashow for sharing information and reagents prior to publication. J.C. is an investigator for the Howard Hughes Medical Institute. This work was supported by National Institutes of Health (NIH) grant R35-GM122604 to J.C. and by the Howard Hughes Medical Institute. A.S. received support from the Salk Institute Pioneer Postdoctoral Endowment Fund. We thank Andrew Gregory, Megan Rae, and Ashnah Shwany for technical assistance. This work was supported by the Next Generation Sequencing Core Facility at the Salk Institute with funding from NIH-National Cancer Institute Cancer Center Support Grant P30 014195, the Chapman Foundation, and the Helmsley Charitable Trust. This publication includes data generated at the University of California, San Diego, Institute of Genetic Medicine Genomics Center utilizing an Illumina NovaSeq 6000 that was purchased with funding from NIH Shared Instrumentation Grant S10 OD026929. This manuscript is dedicated to Dr. Joanne Chory—a fearless explorer, exceptional mentor, and motivating role model.

## Author contributions

A.S. and J.C. conceptualized the study, designed experiments, and wrote the paper. A.S. performed all experiments and analyzed the data. J.C. acquired funding and supervised.

## Competing interests

The authors declare no competing interests.
