## [Transparent Peer Review file · Nature Communications]

Genetic architecture of a light-temperature coincidence detector

Corresponding Author: Dr Adam Seluzicki

Version 0:

Reviewer comments:

Reviewer #1

(Remarks to the Author)

The ms by Seluzicki and Chory report an interesting finding that PHOT2 blue photoreceptor promotes floral initiation in response to ambient low temperature. PHOTs are best known for their role regulating movement responses from the plasma membrane/cytosol. This article reports two novel findings, that PHOT promotes flowering at low temperature, and that PHOT signaling may regulate transcription in the nucleus. The authors provide strong genetic evidence supporting a mechanistic hypothesis explaining the two findings. This report should be interesting to the broad readership of the journal.

Major comment

In the model depicted in Fig. 6, the 4 proteins are grouped together as the “temperature sensor” responsive to the blue light signal perceived by PHOT2. The authors provide abundant genetic evidence supporting this model, but it would be more compelling if they may provide a molecular interpretation. Specifically, the present model may include the experimental evidence that explains the subcellular localization of EHB1 and CAMTA2 and their possible association with the PHOT2/NPH3 complex. Given the previous results that PHOT2 physically interacts with NPH3, NPH3 physically interacts with EHB1, and that EHB1 is a putative lipid-binding protein associated with the function of PHOT2 and/or NPH3, these proteins may physically associate with each other and/or with the plasma membrane, and such physical association may change in response to temperature changes. It seems plausible that CAMTA2 may act like some known transcription factors that shuttle between membrane/cytosol and the nucleus to regulate transcription. Regardless of this speculation, the authors may test the subcellular localization of EHB1 and CAMTA2 and/or whether they physically associate with PHOT2 or NPH3, using any experimental systems, such as the transient *Nicotiana benthamiana* expression system (10.1038/s41467-021-26332-6). A test of the subcellular localization and/or physical association in response to temperature changes would be helpful (although not absolutely necessary at present).

Minor comments

1. Line 89: “Thus, temperature sensitivity unmasked in the *phot2* mutant occurs downstream of, or in parallel with, ELF3 function”. Could the results of the epistasis analysis of *phot2* and *elf3* be interpreted as ELF3 acts downstream to *Phot2*?
2. Line 88: “We found that *elf3* and *phot2 elf3* were identical at 20C” may be changed to “We found that the flowering-time of *elf3* and *phot2 elf3* was identical at 20C”
3. It seems the other allele of *photo2* was isolated from the *Col-gI1* mutant background impaired in the *GLABRA* gene, but it not clear. This may be clarified.
4. It is not very clear exactly what point the authors want to make with respect to the results of analysis presented in Fig. 4. For example, does EHB1 belong to any of the communities 3,6,15? Are genes co-regulated by PHOT2 and CAMTA2 of these communities expected to play roles in the control of light- and/or temperature-dependent regulation of flowering? Some additional analyses may be helpful to clarify this. For example, one may search the literature to clarify how many genes that the mutants or ox lines have been previously reported to show the phenotypic effects on flowering or light or temperature responses. Regardless of the results of this or other type of literature search, such analysis would seem to at least partially address the question what Fig. 4 tried to argue for.
5. Line 201, is *camta2-2* of Fig. 5B “an independent RNA null allele of CAMTA2”? is this *camta2-1* or *camta2-2* allele?

Reviewer #2

(Remarks to the Author)

In this manuscript, the authors showed that the phototropin blue-light photoreceptor is involved in light and low temperature-dependent flowering regulation. Phototropin mutants, especially phot2, show late flowering phenotypes under long-day conditions. This phenotype is more obvious under lower temperatures (ex., 15 degrees C compared to 25 C). They also showed that phototropin-interacting NPH3 is also involved in this regulation. Their omics approach facilitated them in identifying the involvement of CAMTA transcription factors and their likely target gene, EHB1, which is known to work with NPH3, in this flowering regulation. The authors analyzed diurnal transcriptome analysis to investigate potential causes of flowering phenotypes of these mutants. Although finding that phototropin signaling, together with CAMTA TFs, is involved in flowering time regulation under lower ambient temperatures is novel, the authors unfortunately could not connect phototropin signaling or CAMTA to any known flowering time regulation at the molecular level.

Major comments:

- 1) Diurnal transcriptome analysis found some differences in transcription among Col-0, phot2, camta2, and phot2 camta2 mutants. However, it looks like there are no clear connections between these mutants and known flowering genes, as none of the differentially expressed genes in phot1 and camta2 mutants are flowering genes. This might be caused by analyzing transcriptome using 10-day-old plants. Arabidopsis plants flower later in 15 C than in 20 C (Figure 1, etc.). Therefore, the flowering time gene difference may possibly occur later than 10 days after germination. Analyzing gene expression (by either RNA-seq or Q-PCR of known flowering genes) in different growth stages may help to find the potential link of phototropin signaling (through CAMTA TFs) with flowering pathways.
- 2) Figure 1A shows that both phot1 and phot2 showed slight late flowering phenotypes. Although Figure 1B showed that the phot2 phenotype is stronger, since phot1 and phot2 are known to have some redundant function for regulating various phototropin responses, analyzing a double mutant phenotype would be informative. Please include the flowering phenotype of phot1 phot2 double mutants.
- 3) The authors reported that nph3 phenotype is visible under white light with 100 micromol m⁻² s⁻¹ intensity but not under that with 50 micromol m⁻² s⁻¹ intensity (Extended Data 2C). However, to analyze the genetic interaction of phot2 and nph3 (Figure 2D), the experiment was performed under white light (50 micromol m⁻² s⁻¹) conditions. To learn the phot2 and nph3 genetic interactions on this regulation, it is more appropriate to analyze the phenotype under white light conditions with 100 micromol m⁻² s⁻¹ intensity.
- 4) Regarding the results shown in Figure 4C, in addition to communities 3,6, and 15, communities 1, 5, and 11 might also be interesting to discuss. Please include the mean expression patterns of the genes in these communities and analyze gene ontology. Based on scale CPM values, in the community 6 and genes in GO Cell Cycle (Figures 4B and 4G), the mean expression patterns of the genes related to Cell Cycle are highly expressed in 15 C than those in 20 C. Does this mean more cells are divided in plants grown in 15 C than in 20 C?

Minor comments:

- 1) Regarding Extended Data 2, there is a typo in the figure legend. "(D)" should be "(C)." Also, in the figures, there are extra "A" and "B" on the Y-axes of 2A and 2B. Figure C also has a typo. "/" should be inserted between "m²" and "s." (it should be "...mol/m²/s.")
- 2) The legend of Figure 3 did not explain the color difference in Figure 3C. Also, please explain what "DEG frequency" means. What are "0.00, 0.05, and 0.10" in the Figure 3C?
- 3) On line 333, there are no citations for phot mutants.

Reviewer #3

(Remarks to the Author)

This submission from Adam Seluzicki and Joanne Chory investigates molecular processes through which light and temperature signals intersect. It explores how signals from the light receptor phot2 and the transcription factor CATMA2 converge to regulate flowering time under low ambient temperatures. The authors provide evidence that EBH1 provides an integrating link, since its known to be a PHOT2 signaling component, the ebh1 mutant has a flowering phenotype, and its expression is up-regulated in low temperature by CATMA2. The manuscript is well written, experiments are conducted to a high standard with appropriate statistical methods applied. Experimental data backs up many of the claims made but occasionally the evidence presented is relatively weak. Progression through the manuscript is generally logical, but there are some gaps in the data which are highlighted in the comments below.

Comments

1. The impact of phot1 and phot2 on flowering time is relatively subtle. For instance, a temperature reduction from 20C to 15C leads to significant increases in LN at flower bud appearance in the WT. In comparison the impact of phot2 (and phot1) on flowering time is relatively small (see Fig1a). This observation should be reflected in the text description.
2. The subtle phot1 mutant flowering phenotype appears to similar at 20/15C. Is this the case at 25C? Has phot1;phot2 double mutant analysis been conducted? It would be interesting to establish whether combining mutant alleles is additive at 15C.

3. It is unclear why the *gl1* allele is used in this study and whether the data in Fig 1b is directly comparable with Fig1a (is this the same or a separate experiment?). Comment: The *gl1* allele appears to have an accelerating effect on flowering time (compared to WT), particularly at 15C (not mentioned in the results section text). This presumably means GL1 has some role in moderating flowering time. If this is the case then the larger effect observed in *phot2;gl1* double (compared to *phot2* vs Col0) could be (at least in part) due to a *gl1-phot2* genetic interaction. Though of course the general point that *phot2* delays flowering in both genetic backgrounds supports a role for PHOT2 flowering at 20C and 15C.

4. The authors state (p6) “We screened a panel of *camta* mutants in our flowering assay”. It would be helpful to indicate the number and identity of *camta* mutants in the panel.

5. Comment: while the *phot2* and *catma2* alleles have a similar flowering phenotype, RNAseq shows more *phot2* mis-regulated genes at 15C, and more *catma2* misregulated genes at 20C – the majority being down-regulated. This presumably means that overall PHOT2 and CATMA2 have differing temperature-dependent transcriptome responses.

6. Since the focus of this study is flowering, it would be helpful to identify and highlight any temperature-controlled flowering genes which occur in the common DEG sets?

7. EHB1 was identified as a gene that was up-regulated by CATMA2 at the cooler 15C. In vitro EMSA assays provide evidence that this regulation may be through direct binding to the CAMTA (CGCG) motif in the EHB1 promoter. It would be interesting to establish whether binding is also observed in vivo and whether it is temperature dependent.

8. As the authors did not report on whether CATMA2 -promotion of EHB1 expression leads to a concomitant rise in protein levels. Therefore, they cannot claim with certainty that this is the key (cool temperature-activated) step. It would be helpful if this is reflected in the language used.

9. The authors state (p12) “Plants lacking EHB1 or CAMTA2 have identical low-temperature-dependent flowering phenotypes as well, consistent with the hypothesis that EHB1 is the primary factor linking CAMTA2 to PHOT2” I agree it is consistent with the notion, but genetic evidence e.g. *ebh1;camta2*, *ebh1;pch3* and / or *ebh1;phot2* mutants would have been helpful to corroborate the claims.

Minor errors

It would be helpful if the correct notation for each mutant allele is shown in the text and figure legends.

Re: extended data Fig 2. There is an error in Fig. 2a+b, with “A” and “B” appearing in the axes. Fig2C is mislabelled D in the figure legend.

Version 1:

Reviewer comments:

Reviewer #1

(Remarks to the Author)

This reviewer is satisfied with the responses and revisions

Reviewer #2

(Remarks to the Author)

This reviewer appreciated the authors' efforts to address my previous question about the possible molecular link between *phot2*/CAMTA and flowering time. Although we still don't know whether the changes they found contribute to the observed changes in flowering time, seeing changes in the gene expression of some genes that affect flowering is very helpful. The authors also responded successfully to my other comments. With these changes, I am satisfied with the current contents. This manuscript nicely showed that *phot2* is involved in flowering under lower ambient temperatures. I don't have any further questions or comments.

RESPONSE TO REVIEW

"Genetic architecture of a light-temperature coincidence detector"
Adam Seluzicki & Joanne Chory

Reviewer remarks are in italics, headed with ">"
Author responses are indented in plain text.

Reviewer #1 (Remarks to the Author)

>The ms by Seluzicki and Chory report an interesting finding that PHOT2 blue photoreceptor promotes floral initiation in response to ambient low temperature. PHOTs are best known for their role regulating movement responses from the plasma membrane/cytosol. This article reports two novel findings, that PHOT promotes flowering at low temperature, and that PHOT signaling may regulate transcription in the nucleus. The authors provide strong genetic evidence supporting a mechanistic hypothesis explaining the two findings. This report should be interesting to the broad readership of the journal.

We thank the reviewer for recognizing the strengths of our manuscript.

>Major comment

*In the model depicted in Fig. 6, the 4 proteins are grouped together as the "temperature sensor" responsive to the blue light signal perceived by PHOT2. The authors provide abundant genetic evidence supporting this model, but it would be more compelling if they may provide a molecular interpretation. Specifically, the present model may include the experimental evidence that explains the subcellular localization of EHB1 and CAMTA2 and their possible association with the PHOT2/NPH3 complex. Given the previous results that PHOT2 physically interacts with NPH3, NPH3 physically interacts with EHB1, and that EHB1 is a putative lipid-binding protein associated with the function of PHOT2 and/or NPH3, these proteins may physically associate with each other and/or with the plasma membrane, and such physical association may change in response to temperature changes. It seems plausible that CAMTA2 may act like some known transcription factors that shuttle between membrane/cytosol and the nucleus to regulate transcription. Regardless of this speculation, the authors may test the subcellular localization of EHB1 and CAMTA2 and/or whether they physically associate with PHOT2 or NPH3, using any experimental systems, such as the transient *Nicotiana benthamiana* expression system (10.1038/s41467-021-26332-6). A test of the subcellular localization and/or physical association in response to temperature changes would be helpful (although not absolutely necessary at present).*

EHB1 and NPH3 localization were examined using transient expression in tobacco leaves. While clear blue-light-activated mobilization of NPH3 was clear, EHB1 showed an apparent diffuse cytoplasmic localization and was not obviously associated with NPH3 condensates (Extended Data 10). There is potential co-localization in the dark, when NPH3 is near the membrane, although this is not definitive given the constraints of the assay. These new data are consistent with the hypothesis of EHB1 modulating some aspect of NPH3 function at the membrane and are described in the text at lines 288-298. Unfortunately the rapid nature of the NPH3 signaling cycle means that a rigorous examination of temperature effects is not possible with our imaging equipment.

I attempted similar co-localization experiments between EHB1 and two different CAMTA2 constructs. This failed, possibly due to issues with CAMTA2 over-expression under the 35S promoter. EHB1, which expressed very well by itself or with NPH3, was not visible with CAMTA2.

A statement underscoring the unknown biochemical mechanisms underlying this genetic system has been included in the discussion at line 394-395.

> *Minor comments*

> 1. Line 89: “Thus, temperature sensitivity unmasked in the *phot2* mutant occurs downstream of, or in parallel with, *ELF3* function”. Could the results of the epistasis analysis of *phot2* and *elf3* be interpreted as *ELF3* acts downstream to *Phot2*?

Given the inherent complexity and unknowns involved in genetic interactions I cannot categorically rule out that the *ELF3* effect is downstream of *PHOT2*. I observe no *PHOT2*-dependent changes in *ELF3* or *XBAT31* transcripts (*XBAT31* being a regulator of temperature-dependent *ELF3* stability). Genotypes are color coded as in the manuscript, with Col-0 in black, *phot2* (p2) in orange, *camta2* (c2) in blue, and *phot2 camta2* (p2c2) in green.

While transcript data do not exclude possible protein-level interactions, the *elf3* mutant does not completely mask the *phot2* low temperature flowering delay, as would be expected from such a strong mutant if they were acting in a linear pathway. I interpret this interaction as *elf3* setting the early flowering baseline, and *phot2* permitting a slight delay specifically in lower temperature, consistent with *PHOT2* acting downstream or in parallel with *ELF3*. I have softened the language in line 98; “Thus, temperature sensitivity unmasked in the *phot2* mutant most likely occurs downstream of, or in parallel with, *ELF3* function”

> 2. Line 88: “We found that *elf3* and *phot2 elf3* were identical at 20C” may be changed to “We found that the flowering-time of *elf3* and *phot2 elf3* was identical at 20C”

The sentence has been changed.

> 3. It seems the other allele of *photo2* was isolated from the Col-*gl1* mutant background impaired in the *GLABRA* gene, but it not clear. This may be clarified.

PHOT2 was originally identified when the *phot2-1 gl1* mutant was isolated from *gl1* subject to fast neutron bombardment mutagenesis. The origin of this line has been clarified in the Materials and Methods (line 435).

> 4. It is not very clear exactly what point the authors want to make with respect to the results of analysis presented in Fig. 4. For example, does *EHB1* belong to any of the communities 3,6,15? Are genes co-regulated by *PHOT2* and *CAMTA2* of these communities expected to play roles in the control of light- and/or temperature-dependent regulation of flowering? Some additional analyses may be helpful to clarify this. For example, one may search the literature to clarify how many genes that the mutants or ox lines have been previously reported to show the phenotypic effects on flowering or light or temperature responses. Regardless of the results of this or other type of literature search,

such analysis would seem to at least partially address the question what Fig. 4 tried to argue for.

As the Reviewer notes, PHOTs are not generally considered to regulate gene expression. It is therefore necessary and appropriate to present general analysis of these findings. As such, the point of Figure 4 is to describe potential downstream effects of PHOT2/CAMTA2 that may influence growth or development. The observed up-regulation of cell division, translation, and DNA replication systems, in aggregate, is indeed consistent with an increased number of leaves at flowering, although further study is necessary to determine the extent and the effected tissues. Literature searches for genes in these communities did not reveal clear associations with flowering. Perhaps these genes, associated with cell-level properties and phenotypes, have not been carefully tested with respect to an organism-level phenotype such as flowering time. It is also possible that the combined up-regulation of these genes represent some downstream effects of the change in flowering, and thus would not themselves cause a change in flowering when mutated or over-expressed. A statement to this effect has been added to the text at line 243-245.

Obvious changes in the core flowering time genes were not apparent. As noted by Reviewer #3, the flowering time phenotype, while quite consistent, is modest, suggesting several genetic or biochemical steps between the PHOTs and the end result of increased leaf number. Please also see the response to Reviewer #2, major comment 1. Notable temperature- and flowering-related genes identified as differentially expressed are now discussed in the text at lines 181-224. Additional data on known flowering related factors is included in Extended Data 5-7.

> 5. Line 201, *is camta2-2 of Fig. 5B “an independent RNA null allele of CAMTA2”? is this camta2-1 or camta2-2 allele?*

The "independent RNA null allele of CAMTA2" is indeed *camta2-2*. This has been clarified in the text at line 277.

Reviewer #2 (Remarks to the Author)

> In this manuscript, the authors showed that the phototropin blue-light photoreceptor is involved in light and low temperature-dependent flowering regulation. Phototropin mutants, especially phot2, show late flowering phenotypes under long-day conditions. This phenotype is more obvious under lower temperatures (ex., 15 degrees C compared to 25 C). They also showed that phototropin-interacting NPH3 is also involved in this regulation. Their omics approach facilitated them in identifying the involvement of CAMTA transcription factors and their likely target gene, EHB1, which is known to work with NPH3, in this flowering regulation. The authors analyzed diurnal transcriptome analysis to investigate potential causes of flowering phenotypes of these mutants. Although finding that phototropin signaling, together with CAMTA TFs, is involved in flowering time regulation under lower ambient temperatures is novel, the authors unfortunately could not connect phototropin signaling or CAMTA to any known flowering time regulation at the molecular level.

We agree that the connection to canonical flowering pathways remains a significant open question from this study, and we have now included substantial new data addressing this issue (see response to Major Comment 1 below).

> *Major comments:*

> 1) *Diurnal transcriptome analysis found some differences in transcription among Col-0, phot2, camta2, and phot2 camta2 mutants. However, it looks like there are no clear connections between these mutants and known flowering genes, as none of the differentially expressed genes in phot1 and camta2 mutants are flowering genes. This might be caused by analyzing transcriptome using 10-day-old plants. Arabidopsis plants flower later in 15 C than in 20 C (Figure 1, etc.). Therefore, the flowering time gene difference may possibly occur later than 10 days after germination. Analyzing gene*

expression (by either RNA-seq or Q-PCR of known flowering genes) in different growth stages may help to find the potential link of phototropin signaling (through CAMTA TFs) with flowering pathways.

A few flowering-related genes were identified as differentially expressed in the *phot2* and *camta2* mutants, and a description of some of these is included at lines 181-190.

Expression of two of two less-well-characterized members of the *FT* family, *TSF* (a floral promoter) and *ATC* (a floral repressor), may be altered in the *phot2* and *camta2* mutants in a manner consistent with a delay in flowering, and consistent with the magnitude of the flowering phenotype. Newly included figures show RNA-seq (Extended Data 5) and qRT-PCR data (Extended Data 6) for these genes. These new data are described at lines 191-206.

Three known regulators of the floral transition, *SOC1*, *SVP*, and *SEP3*, have now been examined by qRT-PCR at ZT8 (peak expression time for these three genes) on day 15 at 15C and 25C, and are included in Extended Data 7. *SOC1* and *SVP* are best characterized as acting in a feedback loop with *FT* to integrate environmental inputs, and *SEP3* integrates multiple inputs to control floral determination in the switch from vegetative to reproductive growth. We observe a trend of reduced *SEP3* transcript in *phot2* and *camta2* mutants at 15C, consistent with a later switch to reproductive growth. Detailed examination of the transition from vegetative to reproductive meristem will require more restricted sampling, as the whole-shoot samples used here may potentially dilute some tissue-specific effects. The need for further examination into the specific signal transduction links between PHOTs/NPH3/CAMTA2 and flowering control pathways at higher temporal and spatial resolution is discussed (lines 220-224 and 263-266).

> 2) Figure 1A shows that both phot1 and phot2 showed slight late flowering phenotypes. Although Figure 1B showed that the phot2 phenotype is stronger, since phot1 and phot2 are known to have some redundant function for regulating various phototropin responses, analyzing a double mutant phenotype would be informative. Please include the flowering phenotype of phot1 phot2 double mutants.

Flowering time data for the *phot1 phot2* double mutant constructed from the SALK lines is included in Extended Data 2D and described at lines 83-90. This figure shows the double mutant is similar to the *phot2* single mutant, while *phot1* is similar to Col-0.

> 3) The authors reported that nph3 phenotype is visible under white light with 100 micromol m-2 s-1 intensity but not under that with 50 micromol m-2 s-1 intensity (Extended Data 2C). However, to analyze the genetic interaction of phot2 and nph3 (Figure 2D), the experiment was performed under white light (50 micromol m-2 s-1) conditions. To learn the phot2 and nph3 genetic interactions on this regulation, it is more appropriate to analyze the phenotype under white light conditions with 100 micromol m-2 s-1 intensity.

We have included data for *phot2 nph3* under 100umol m⁻²s⁻¹ in Extended Data 2D and described at line 120-121. The genetic relationship between *phot2* and *nph3* is preserved and continues to support our hypothesis of NPH3 acting downstream of, and being required for, PHOT2 regulation of flowering. In this experiment a trend of *nph3* early flowering is preserved, but not statistically significant at 15C, possibly due to the presence of an outlier in *nph3*. I have therefore softened the language describing light intensity-dependent early flowering and highlighted it as a direction for further study (line 126).

> 4) Regarding the results shown in Figure 4C, in addition to communities 3,6, and 15, communities 1, 5, and 11 might also be interesting to discuss. Please include the mean expression patterns of the genes in these communities and analyze gene ontology.

We have included expression and GO data for communities 1 and 11 in Extended Data 9. Description of these communities is added at line 249-262. The figure originally presented as Extended Data 9 (replicate uncropped EMSA images) has been moved to Source Data.

Community 5 is enriched for GO terms associated with response to bacteria and defense (see below). Given that this experiment was carried out using soil-grown plants, it is possible that some plants were responding to the microbiome more strongly than others. Closer inspection of community 5 indicates very high variability among replicates at some time points (see below for representative gene AT1G33960 - AVRRPT2-INDUCED GENE 1 (AIG1)). In contrast, representative genes from communities 3, 6, and 15 show tight distributions (see Extended Data 8). Due to this noise level, I have decided not to include community 5 as a community of interest.

> Based on scale CPM values, in the community 6 and genes in GO Cell Cycle (Figures 4B and 4G), the mean expression patterns of the genes related to Cell Cycle are highly expressed in 15 C than those in 20 C. Does this mean more cells are divided in plants grown in 15 C than in 20 C?

We believe that increased cell division in low temperature is a reasonable hypothesis to draw from these data, and now mention this possibility in the discussion (lines 416-418).

> Minor comments:

> 1) Regarding Extended Data 2, there is a typo in the figure legend. “(D)” should be “(C).” Also, in the figures, there are extra “A” and “B” on the Y-axes of 2A and 2B. Figure C also has a typo. “f” should be inserted between “m2” and “s.” (it should be “..mol/m2/s.”)

We have fixed these typographical errors.

The notation has been adjusted throughout the manuscript to read $\mu\text{mol m}^{-2}\text{s}^{-1}$.

2) The legend of Figure 3 did not explain the color difference in Figure 3C. Also, please explain what “DEG frequency” means. What are “0.00, 0.05, and 0.10” in the Figure 3C?

The plots in 3C are histograms, with the x-axis (time in ZT) on a circle, and the y-axis on the radius, with the center point of the circle being 0. “DEG Frequency” is the proportion of total

DEGs identified in each genotype at each time point. The figure and corresponding legend have been edited to clarify the labels and axes.

3) On line 333, there are no citations for phot mutants.

The necessary citations have been added in the methods.

Reviewer #3 (Remarks to the Author):

> This submission from Adam Seluzicki and Joanne Chory investigates molecular processes through which light and temperature signals intersect. It explores how signals from the light receptor phot2 and the transcription factor CATMA2 converge to regulate flowering time under low ambient temperatures. The authors provide evidence that EBH1 provides an integrating link, since its known to be a PHOT2 signaling component, the ebh1 mutant has a flowering phenotype, and its expression is up-regulated in low temperature by CATMA2. The manuscript is well written, experiments are conducted to a high standard with appropriate statistical methods applied. Experimental data backs up many of the claims made but occasionally the evidence presented is relatively weak. Progression through the manuscript is generally logical, but there are some gaps in the data which are highlighted in the comments below.

Thanks to the reviewer for recognizing the strengths of the manuscript and the potential for improvement.

> Comments

> 1. The impact of phot1 and phot2 on flowering time is relatively subtle. For instance, a temperature reduction from 20C to 15C leads to significant increases in LN at flower bud appearance in the WT. In comparison the impact of phot2 (and phot1) on flowering time is relatively small (see Fig1a). This observation should be reflected in the text description.

The effect size is more clearly noted in the text at line 76.

> 2. The subtle phot1 mutant flowering phenotype appears to similar at 20/15C. Is this the case at 25C? Has phot1;phot2 double mutant analysis been conducted? It would be interesting to establish whether combining mutant alleles is additive at 15C.

Please see response to Reviewer #2, major comment #2. An experiment showing the phot1 phot2 double mutant constructed from the SALK lines been included in Extended Data 2D.

> 3. It is unclear why the gl1 allele is used in this study and whether the data in Fig 1b is directly comparable with Fig1a (is this the same or a separate experiment?). Comment: The gl1 allele appears to have an accelerating effect on flowering time (compared to WT), particularly at 15C (not mentioned in the results section text). This presumably means GL1 has some role in moderating flowering time. If this is the case then the larger effect observed in phot2;gl1 double (compared to phot2 vs Col0) could be (at least in part) due to a gl1-phot2 genetic interaction. Though of course the general point that phot2 delays flowering in both genetic backgrounds supports a role for PHOT2 flowering at 20C and 15C.

The gl1 mutation is in the background of the original phot1 and phot2 mutants, and the relevant background control (gl1 alone) is included in our data for comparison. The potential flowering time difference between gl1 and Col-0 has been noted but the data presented in figure 1 are separate experiments and not

directly comparable. As the reviewer notes, *phot2* loss of function delays flowering in both backgrounds. For the reviewer's reference, please see the figure at right for an experiment comparing *phot2-1* with and without *gl1*.

> 4. The authors state (p6) “We screened a panel of *camta* mutants in our flowering assay”. It would be helpful to indicate the number and identity of *camta* mutants in the panel.

The *camta* family mutants and combinations are presented in figure 3A, and the specific insertions are identified in the corresponding figure legend, and in methods (lines 437-438)

> 5. Comment: while the *phot2* and *camta2* alleles have a similar flowering phenotype, RNAseq shows more *phot2* mis-regulated genes at 15C, and more *camta2* misregulated genes at 20C – the majority being down-regulated. This presumably means that overall PHOT2 and CATMA2 have differing temperature-dependent transcriptome responses.

We agree that the effects of these mutants on the transcriptome are not completely overlapping, and have now described them as such in the text (see line #407-409).

> 6. Since the focus of this study is flowering, it would be helpful to identify and highlight any temperature-controlled flowering genes which occur in the common DEG sets?

Known flowering time genes are more clearly noted in the text (line 181-190). In short, few of the classical flowering time genes are identified as DEGs. Please see the response to Reviewer #2, major comment 1 for a description of additional data (added to the paper in Extended Data 5-7).

> 7. *EHB1* was identified as a gene that was up-regulated by *CATMA2* at the cooler 15C. In vitro EMSA assays provide evidence that this regulation may be through direct binding to the *CAMTA* (CGCG) motif in the *EHB1* promoter. It would be interesting to establish whether binding is also observed in vivo and whether it is temperature dependent.

I completely agree that temperature-dependent promoter occupancy of *CAMTA2* *in vivo* is an interesting and important question, particularly with respect to regulation by environmental factors. Unfortunately, tagged *CAMTA2* transgenic *Arabidopsis* lines that might be used for ChIP are not currently available, and I am not aware of any validated antibodies to native *CAMTA2*.

Examination of *CAMTA2*'s ability to activate an *EHB1* promoter-driven luciferase reporter using tobacco leaf infiltration was attempted but failed due to technical problems. These hurdles are being addressed, but unfortunately this issue has not yet been resolved. I have added text highlighting the future need for an *in vivo* binding assay in the text at lines 285-286.

> 8. As the authors did not report on whether *CATMA2* -promotion of *EHB1* expression leads to a concomitant rise in protein levels. Therefore, they cannot claim with certainty that this is the key (cool temperature-activated) step. It would be helpful if this is reflected in the language used.

An appropriate note has been included in the text (line #388).

> 9. The authors state (p12) “Plants lacking *EHB1* or *CAMTA2* have identical low-temperature-dependent flowering phenotypes as well, consistent with the hypothesis that *EHB1* is the primary factor linking *CAMTA2* to *PHOT2*” I agree it is consistent with the notion, but genetic evidence e.g. *ebh1;camta2*, *ebh1;pch3* and / or *ebh1;phot2* mutants would have been helpful to corroborate the

claims.

The reviewer raises a very good point, and the behavior of the proposed combinations is definitely of interest. While line construction for some of these double mutants is in progress, I do not yet have them available and look forward to examining them in the future. I have mentioned this in the text at line 395-396.

> *Minor errors*

> *It would be helpful if the correct notation for each mutant allele is shown in the text and figure legends.*

Mutant line annotations (T-DNA line IDs) previously included in Methods are now also included in text and figure legends.

> *Re: extended data Fig 2. There is an error in Fig. 2a+b, with "A" and "B" appearing in the axes. Fig2C is mislabelled D in the figure legend.*

These errors have been corrected.